# Adsorption of Sb (III) on Oxidized Exfoliated Graphite Nanoplatelets

**DOI:** 10.3390/nano8120992

**Published:** 2018-11-30

**Authors:** Luiza Capra, Mihaela Manolache, Ion Ion, Rusandica Stoica, Gabriela Stinga, Sanda Maria Doncea, Elvira Alexandrescu, Raluca Somoghi, Marian Romeo Calin, Ileana Radulescu, Georgeta Ramona Ivan, Marian Deaconu, Alina Catrinel Ion

**Affiliations:** 1National Research & Development Institute for Chemistry and Petrochemistry ICECHIM, 202 Splaiul Independentei Street, 060021 Bucharest, Romania; dir.calitate@icechim.ro (M.M.); irusandica@yahoo.com (R.S.); sandamariadoncea@gmail.com (S.M.D.); elvira.alexandrescu@icechim-pd.ro (E.A.); r.somoghi@gmail.com (R.S.); rateageorgeta@yahoo.com (G.R.I.); deaconu319@yahoo.com (M.D.); 2Faculty of Applied Chemistry and Material Science, University Politehnica of Bucharest, 313 Splaiul Independentei Street, 060042 Bucharest, Romania; i_ion2000@yahoo.com; 3“Ilie Murgulescu” Institute of Physical Chemistry of the Romanian Academy, 202 Splaiul Independentei Street, 060021 Bucharest, Romania; gstinga@gmail.com; 4“Horia Hulubei” National Institute for Physics and Nuclear Engineering—IFIN HH, 30 Reactorului Street, P.O. Box MG-6, 077125 Bucharest-Magurele, Romania; rcalin@nipne.ro (M.R.C.); rileana@nipne.ro (I.R.)

**Keywords:** Sb(III), adsorption, oxidized exfoliated graphite nanoplatelets

## Abstract

In this work, Sb (III) adsorption on oxidized exfoliated graphite nanoplatelets (ox-xGnP) was evaluated for the first time, to the best of our knowledge. The ox-xGnP were characterized by thermogravimetric analysis (TGA), Fourier transform infrared spectroscopy (FT-IR), Brunauer–Emmet–Teller (BET) analysis, scanning electron microscopy (SEM), transmission electron microscopy (TEM) equipped with energy-dispersive X-ray spectroscopy (EDX), and Zeta potential analysis. The adsorption parameters, such as pH and contact time, were optimized, and the best adsorption capacity obtained was 8.91 mg g^−1^ at pH = 7.0, 1.0 mg ox-xGnP/100 mL solution, T = 293 K, 1.0 mg L^−1^, Sb (III), 25 min contact time. The best correlation of the kinetic data was described by a pseudo-first-order kinetic model, with R^2^ = 0.999. The adsorption isotherms of Sb (III) onto ox-xGnP were best described by the Langmuir isotherm model. The thermodynamic parameters showed that the adsorption process was exothermic and spontaneous.

## 1. Introduction

Drinking water is a vital element for human life and health. Therefore, it is very important for it to be devoid of heavy metals which can cause various diseases in human body [1]. Antimony (Sb) is widespread in the natural environment and is a toxic element even at very low concentrations. The compounds of Sb (III) are 10 times more toxic than those of Sb (V) [2]. Sb can derive from natural processes as well as from human activities [3]. Taking into consideration the toxicity and the effects induced by antimony, many methods have been developed to remove this element from various environments in recent years. Hence, a high number of research studies have demonstrated that antimony can be eliminated from aqueous solutions by reverse osmosis [4], solvent extraction [5], reduction and precipitation [6], coagulation [7], electrodeposition [8], ion exchange [9], filter membranes [10], biosorption [11,12], and adsorption [13]. Among these, the sorption method was considered one of the best alternatives for removing Sb from aqueous solutions because of its low cost, simplicity, rapidity, lack of toxicity, low energy consumption, high efficiency, and the possibility to regenerate the absorbent. Currently, the adsorbents used to remove Sb (III) are natural materials such as perlite, bentonite, hydroxyapatite, metal oxides (like the Fe–Mn binary oxide), activated alumina, and carbon-based materials (such as multi-walled carbon nanotubes MWCNTs, graphene, active carbon) [14].

Few studies on the adsorption of Sb (III) on carbon nanomaterials are found in the literature [15,16,17]; therefore, the purpose of this study was to evaluate the potential of oxidized exfoliated graphite nanoplatelets (ox-xGnP) as adsorbents for Sb (III). To the best of our knowledge, it is the first time that ox-xGnP are applied as adsorbents for Sb (III). This adsorbent was selected for the present study because exfoliated graphite nanoplatelets (xGnP) represent [18] a new effective nanolayered sorbent material with high adsorption capacity, good stability, and rapid adsorption [19]. In addition, xGnP could be a suitable substituent for carbon nanotubes and fullerenes because of their low cost [20]. The effect of several parameters, such as pH, amount of ox-xGnP, contact time, initial concentration of Sb (III), and temperature, was studied. The kinetics and the adsorption isotherms of the process were studied to achieve the best kinetic model and the best isotherm equation, respectively. In addition, the thermodynamic parameters were studied.

## 2. Materials and Methods

### 2.1. Materials

xGnP were purchased as a powder from XG Sciences, Inc., (Lansing, MI, USA); H_2_SO_4_ 98% and HNO_3_ 65% were purchased from Fluka (Göteborg, Sweden) and were used to oxidize xGnP. Potassium tartrate and antimony (III), >99.9% purity, were from Merck Group (Darmstadt, Germany). To adjust the pH, standard traceable 0.1 M KOH and 0.1 M HCl solutions Scharlau (Barcelona, Spain) were used. A standard solution, Quality Control Standard 21, 100 mg L^−1^ Sb concentration, from Perkin Elmer (Waltham, MA, USA) was used to plot the calibration curve for inductively coupled plasma optical emission spectrometry (ICP–OES). For the preparation of work solutions and standards, ultrapure water with resistivity 18.2 MΩ cm^−1^, produced by an EASY pure RoDi equipment (Barnstead, NH, USA) was used. All work solutions were freshly prepared before use. The purge gas for ICP–OES was Argon 5.0, >99.999% purity (Linde Gaz, Romania).

### 2.2. Synthesis of ox-xGnP and Characterization Methods

The xGnP (black powder) was functionalized by chemical oxidation to obtain ox-xGnP, according to the method presented in references [21,22]. For the characterization of ox-xGnP before and after adsorption, the following methods and equipment were used:

Thermogravimetric analysis (TGA) was performed using the Mettler Toledo (Columbus, OH, USA) TGA/SDTA 851 apparatus in nitrogen atmosphere with alumina crucibles in the temperature range of 50–550 °C and a heating rate of 1 °C min^−1^. The Brunauer–Emmet–Teller (BET) specific surface area of the ox-xGnP was characterized by nitrogen adsorption using a Quantachrome NOVA 2200e instrument (Boynton Beach, FL, USA). Nitrogen adsorption–desorption isotherm was measured at the temperature of liquid nitrogen (77 K). Prior to measurements, the samples were degassed at 60 °C in vacuum for 4 h. The specific surface area was calculated according to the BET equation. The total pore volume was estimated from the amount of gas adsorbed at a relative pressure p/p_o_ = 0.99. The classical Barrett–Joyner–Halenda (BJH) model applied to the adsorption branch of the isotherm was used to determine mesopores’ surface area, mesopores’ volume, and mesopores’ size distribution, respectively [23] Zeta potential was obtained from the electrophoretic mobility by using the Smoluchowsky model. The electrokinetic measurements were carried out with the Zetasizer Nano ZS, Malvern Instruments Ltd., UK. The ζ-potential can be used as an indicator of the electrostatic stabilization of particles. The electrokinetic measurements were performed on ox-xGnP at different pH values. Fourier transform infrared spectroscopy (FT-IR): spectra were recorded on a Spectrum GX spectrometer Perkin Elmer (Waltham, MA, USA) in transmittance, in KBr pellets, by accumulation of 32 spectra, with a resolution of 4 cm^−1^. Scanning Electron microscopy (SEM): the images were obtained with a FEI QUANTA 200 (FEI Company, Hillsboro, OR, USA) equipment operating at a 30 kV electron acceleration voltage, high vacuum, direct on uncovered samples. The microscope Tecnai™ G2 F20 TWIN Cryo-TEM, 2015 (FEI Company, Eindhoven, The Netherlands) was used to perform classical TEM (BF-TEM) and scanning transmission electron microscopy (STEM) analyses on final materials. The powders were well dispersed in ultrapure water by ultrasonic treatment of the samples. EDX (single point) was also used to determine the presence of different elements. A small drop of well-dispersed sample was placed on the copper grid and visualized by TEM.

### 2.3. Description Of the Adsorption Experiments

A stock solution of 100 mg L^−1^ Sb (III) used in the adsorption experiments was prepared by dissolving 0.137 g of potassium antimony tartrate in a 500 mL volumetric flask with ultrapure water. From the stock solution, work solutions with the concentrations of 0.1 mg L^−1^, 0.3 mg L^−1^, 0.5 mg L^−1^, 0.7 mg L^−1^, and 1.0 mg L^−1^ were prepared by dilution. The pH of the working solutions was adjusted by adding small amounts of 0.1 or 0.01 M KOH solution and 0.1 M HCl. For the study of the adsorption kinetics, solutions were prepared by adding 100 mL of solution adjusted to pH = 4.0, 5.0, 7.0, 9.0, and 11.0 over 1.0 mg of ox-xGnP. After ultrasonic treatment of ox-xGnP for 30 min, adequate volumes of Sb (III) stock solution to obtain the concentrations of 0.1 mg L^−1^, 0.3 mg L^−1^, 0.5 mg L^−1^, 0.7 mg L^−1^, and 1.0 mg L^−1^ were added.

The dispersion of -ox-xGnP was performed using an Ultrasonic Processor VCX 750 (Newtown, CT, USA) with a frequency of 20 kHz and an ultrasonic bath Elma P-30H Ultrasonic (Landsberger, Berlin) operated at a frequency of 37 Hz at 20 °C, 25 °C, and 30 °C. After completion of the adsorption experiments, the suspensions were filtered into 2.0 mL glass vials using a 0.2 μm RC filter attached to a syringe, and the concentration of Sb was analyzed by the ICP–OES Optima 2100 DV System (Perkin Elmer, Waltham, MA, USA) with dual optics view. The operating conditions of the equipment are described in a previous paper [24].

All experimental data were obtained in triplicate, and the relative standard deviations were <5%. The models were calculated on the basis of the average of the values obtained for each point.

The adsorption capacity *q_t_* (mg g^−1^) was calculated by Equation (1):(1)$$q_{t}=\frac{\left(C_{0}-C_{x}\right)}{m}\times V$$
where *C*_0_ (mg L^−1^) and *C_x_* (mg L^−1^) are the initial concentration of Sb (III) in solution and the concentration at time *t*, respectively; *V* (L) is the volume of solution; *m* (g) is the mass of ox-xGnP.

## 3. Results

### 3.1. Characterization of ox-xGnP

The TGA curve of the ox-xGnP is shown in Figure 1. From the obtained thermogram, a very small rise of the mass (%) of about 1% can be observed between 50 and 130 °C, which might be be explained by the adsorption of N_2_ on the xGnP surface, but the major change observed from the thermogram is a decrease of the mass loss of 4.1% between 130 and 550 °C, which can be attributed to the release of CO and CO_2_, according to the literature [25].

The following results from the N_2_ physisorption analysis of ox-xGnP were obtained: mesopores’ surface area = 205 m^2^ g^−1^, mesopores’ volume = 0.33 cc g^−1^, and mesopores’ diameter = 3.9 nm, respectively. Figure 2 compares the FT-IR spectrum of ox-xGnP with that of ox-xGnP on which Sb (III) from antimony potassium tartrate was adsorbed, (ox-xGnP+Sb). In this study, the spectral zone from 1800 cm^−1^ to 1000 cm^−1^ was analyzed.

The occurrence of Sb (III) adsorption was highlighted by the disappearance of the 1745 cm^−1^ vibration band, specific to the C=O group, from the oxidized graphite and the appearance of vibration bands for –COO at 1704 cm^−1^, 1688 cm^−1^, 1564 cm^−1^, 1318 cm^−1^, and 1282 cm^−1^. The vibration bands specific to the C–O–C group at 1260 cm^−1^, 1247 cm^−1^, 1108 cm^−1^ and that specific to the C(O)–O group at 1051 cm^−1^ [26] were observed both before and after adsorption. Additionally, the absorption bands at 1517 cm^−1^, 1507 cm^−1^, 1494 cm^−1^, and 1164 cm^−1^, specific for the –CH groups, were observed after the adsorption process [27,28,29,30].

The morphological study of the oxidized nanomaterials ox-xGnP and ox-xGnP+Sb was performed by scanning electron microscopy. The SEM images of secondary electrons for ox-xGnP with a magnification of 10,000× and 15,000× are presented in Figure 3a,c, respectively. A morphology of extremely thin films was observed, with a slight tendency of crowding and marginal twisting and the presence of bright, rounded, and polyhedral particles of Sb with relatively uniform layout on the samples after the sorption of Sb (Figure 3b,d). The SEM images confirmed the adsorption of Sb on ox-xGnP.

In Figure 4a, a TEM image of ox-GnP before adsorption of Sb is shown, whereas in Figure 4b, a STEM image of ox-GnP after adsorption of Sb is shown. Both micrographs show a mostly agglomerated structure, the first one having a wrinkled paper morphology due to the initial xGnP morphology [31]. The STEM image presents Sb particles having a non-regular shape (as in SEM images), with sizes between 50 and 200 nm. The ox-xGnP sample did not present the wrinkled paper morphology. In Figure 4c, EDX indicated the presence of small quantities of Sb.

### 3.2. Adsorption Study. Effects of the Adsorption Parameters

In order to optimize the adsorption of Sb (III) on ox-xGnP from aqueous solution, the effects of the following parameters were investigated: the pH of the sorption solutions, the amount of ox-xGnP, the contact time and the temperature at which the adsorption process took place, and the initial concentration of Sb (III).

#### 3.2.1. The Effect of pH on the Adsorption Capacity of Sb (III) from Aqueous Solutions Studied in the pH Range of 4.0–11.0

In the range of pH 5.0–9.0, a significant increase in the adsorption capacity of Sb (III) on the ox-xGnP surface, from 7.40 to 16.0 mg g^−1^, was observed; then, the adsorption decreased, reaching a value of 9.60 mg g^−1^ at pH 11.0, as shown in Figure 5. The influence of the solution pH on the adsorption capacity is a complex phenomenon influenced by both the metal ion species in the solution and the modifications of the functional groups on the surface of the adsorbent material. The negative charge of the surface ox-xGnP confirmed by Zeta’s potential analysis (Figure 6) provided favorable interactions for the adsorption of cationic species. By increasing the pH, the values of ζ-potential varied from −13.7 mV (pH = 4.0) to −63.4 mV (pH = 9.0). The higher the value of pH, the more negative the net charge of ox-xGnP; at pH 9.0, the highest colloidal stability due to the electrostatic repulsion between particles was observed.

In the pH range between 4.0 and 10.0, the species of Sb, (III) according to the literature are [HSbO_2_] and Sb(OH)_3_ in aqueous solution_._ At pH values > 10.0 the presence of the species Sb(OH)_4_^−^ and [SbO_2_]^−^ was previously mentioned [32,33]. The experiments were done at pH 7.0, in view of further applications of these results for Sb (III) removal from drinking water, the common pH range of which is between 6.0 and 8.0.

#### 3.2.2. Effect of the Amount of ox-xGnP on the Adsorption Process

To evaluate the influence of ox-xGnP amount on adsorption, the sorption experiments were performed at pH = 7.0, T = 293 K, Sb concentrations of 0.3 mg L^−1^ and 0.7 mg L^−1^, and amounts of ox-xGnP 1.0 and 2.0 mg. The data presented in Table 1 show a lower adsorption capacity with increasing adsorbent amounts. Therefore, in the following experiments, 1.0 mg of ox-xGnP was used.

#### 3.2.3. Effect of the Initial Concentration of Sb (III) on the Adsorption Process

Figure 7 shows the variation of the adsorption capacity with the contact time for an amount of ox-xGnP corresponding to 1.0 mg and initial concentrations of Sb of 0.1, 0.3, 0.5, and 1.0 mg L^−1^. The adsorption capacity increased by increasing the initial concentration of Sb (III). Stationary (steady) values of adsorption capacity were reached in all experiments after about 25 min.

#### 3.2.4. Effect of Contact Time and Temperature on the Adsorption Process

The effect of contact time and temperature on the adsorption process was studied for systems containing 1.0 mg of ox-xGnP in solutions of Sb (III) with initial concentration of 1.0 mg L^−1^, at pH 7.0 and Temperatures of 20 °C, 25 °C, and 30 °C, as shown in Figure 8.

An initial increase of the adsorption capacity (qt) of Sb (III) was observed during the first 20 min, after which it remained constant. By increasing the temperature, the values of the adsorption capacity decreased. The time required to reach the equilibrium in the temperature range 20–30 °C varied between 25 and 40 min.

Considering the parameters variation, the optimal working conditions were the following: pH = 7.0, 1.0 mg ox-xGnP/100 mL solution, T = 293 K, 1.0 mg L^−1^ Sb (III), 25 min contact time.

### 3.3. Adsorption Kinetics Study

The experimental data obtained in this study were evaluated on the basis of three kinetic models, namely, the pseudo-first-order kinetic model, the pseudo-second-order kinetic model, and the intra-particle diffusion, to investigate the kinetics of the adsorption mechanism.

The equation of the pseudo-first-order kinetic model, given by Lagergren [34], is written as follows:(2)$$\ln(q_{e}-q_{t})=\ln q_{e}-k_{1}t$$
where, *q_t_* is the amount of Sb (III) adsorbed on ox-xGnP at time *t* (mg g^−1^), *q_e_* is the amount of Sb (III) adsorbed on ox-xGnP at equilibrium (mg g^−1^), *k*_1_ is the rate constant of the pseudo-first-order equation (min^−1^), and *t* is the contact time (min). The rate constant *k*_1_ was obtained from the representation ln (qe − qt) versus t.

The data obtained for the adsorption of Sb (III) on ox-xGnP were also analyzed by the pseudo-second order of kinetics, given by the following equation [35,36]:(3)$$\frac{t}{q_{t}}=\frac{1}{k_{2}q_{e}^{2}}+\frac{t}{q_{e}}$$
where, *q_t_* is the amount of Sb (III) adsorbed on ox-xGnP at time *t* (mg g^−1^), *q_e_* is the amount of Sb (III) adsorbed on ox-xGnP at equilibrium (mg g^−1^), *k*_2_ is the rate constant of the pseudo-second-order equation (g mg^−1^min^−1^), and *t* is the contact time (min). The rate constant *k*_2_ was obtained from the dependence *t*/*qt* versus *t*.

The intra-particle diffusion model [37] is given by the following equation:(4)$$q_{t}=k_{id}t^{1/2}+C$$
where, *q_t_* is the amount of Sb (III) adsorbed on ox-xGnP at time *t* (mg g^−1^), *k_id_* is the intra-particle diffusion constant (mg g^−1^min^−1/2^), *t* is the contact time (min^1/2^), and *C* (mg g^−1^) is a constant, proportional to the thickness of the boundary layer. The constant *k_i_* is obtained from the representation *q_t_* versus *t*^1/2^. The criterion for assessing the applicability of the three kinetic models was the determination coefficient presented in Table 2.

The highest value of R^2^ was obtained with the pseudo-first kinetic model. Moreover, the value of the adsorption capacity obtained experimentally at equilibrium was in accordance with the value of adsorption capacity calculated by the pseudo-first kinetic model (Table 2), which indicated that the adsorption process of Sb (III) on ox-xGnP fit better with this model.

### 3.4. Study of Adsorption Isotherms

The adsorption isotherms describe how the adsorbed Sb (III) interacts with the ox-xGnP, providing information about the nature of the interactions that occur in the process. To describe these processes, the Langmuir and Freundlich models, the equations of which are presented below, were used in this study.

The Langmuir model is a model based on monolayer adsorption and surface homogeneity, the equation [38] of which is:(5)$$q_{e}=\frac{q_{m}K_{L}C_{e}}{1+K_{L}C_{e}}$$
where: *C_e_* is the concentration of Sb (III) at equilibrium (mg L^−1^), *q_e_* is the amount of Sb (III) adsorbed on ox-xGnP at equilibrium (mg g^−1^), *q_m_* is the maximum adsorption capacity in monolayer at equilibrium, *K_L_* is the Langmuir constant (L mg^−1^*)*. The value of *q_m_* and *K_L_* are calculated from the slope and the intercept of the linear equation of *C_e_*/*q_e_* versus *C_e_*. Another important parameter calculated from the Langmuir isotherm is the separation factor constant, *R_L_*, which is given by Equation (6):(6)$$R_{L}=1/\left(1+K_{L}C_{0}\right)$$
where: *K_L_* (L mg^−1^) is the Langmuir constant, and *C*_0_ (mg L^−1^) is the initial Sb (III) concentration.

The R_L_ value provides information about the adsorption process as follows: when *R_L_* > 1, the process is unfavorable, when *R_L_* = 1, it is linear, when (0 < *R_L_* < 1), the process is favorable and when *R_L_* = 0 the process is irreversible [39].

The Freundlich model is an empirical model based on multilayer adsorption and surface heterogeneity, the equation [40] of which is presented below:(7)$$q_{e}=K_{F}C_{e}^{1/n}$$
where: *C_e_* is the concentration of Sb (III) at equilibrium (mg L^−1^), *q_e_* is the amount of Sb (III) adsorbed at equilibrium (mg g^−1^), *K_F_* is the Freundlich constant.

The magnitude of 1/*n* quantifies the favorability of the adsorption process and the surface heterogeneity. *K_F_* and n are obtained from the intercept and slope of the linear equation of *q_e_* versus *C_e_*. The parameters calculated from the Freundlich and Langmuir models at three different temperatures are shown in Table 3.

Figure 9 shows the Freundlich and Langmuir adsorption isotherms of Sb (III) on ox-xGnP at temperatures of 293 K, 298 K, and 303 K.

The criterion for selecting the isotherm that best fits the experimental data is R^2^. The values presented in Table 3 indicate that the Langmuir model best fit the experimental data at isothermal equilibrium. According to this model, the adsorption of Sb (III) on ox-xGnP was performed on homogeneous surfaces in monolayer. The maximum adsorption capacity at 293 K calculated from the Langmuir isotherm was *q_m_* = 18.2 mg g^−1^ versus *q_exp_* = 8.91 mg g^−1^. The *R_L_* values obtained for the three temperatures presented in Table 3 were between 0.42 and 0.49, which confirms that the adsorption of Sb (III) on ox-xGnP is favorable.

### 3.5. Thermodynamic Study

In this study, the thermodynamic parameters associated with the adsorption process were also calculated, giving information about the type of interactions taking place. The thermodynamic parameters, namely, Gibb’s free energy (Δ*G*^0^, kJ mol^−1^), enthalpy (Δ*H*^0^, kJ mol^−1^), and entropy (Δ*S*^0^, J mol^−1^K^−1^), could provide data on the size of the adsorption process changing the internal energy throughout it. The parameters were determined by the following equations [41]:(8)$$\Delta G^{0}=\Delta H^{0}-T\Delta S^{0}$$
(9)$$\Delta G^{0}=-RT\ln K_{L}$$

Based on Equations (8) and (9), Equations (10) and (11) are obtained:(10)$$\Delta H^{0}-T\Delta S^{0}=-RT\ln K_{L}$$
(11)$$\ln\left(K_{L}\right)=\frac{\Delta S^{0}}{R}-\frac{\Delta H^{0}}{R}\cdot \frac{1}{T}$$
where *K_L_* is the Langmuir equilibrium constant (L mol^−1^), *R* is the gas constant (8.314 × 10^−3^ kJ mol^−1^), *T* is the absolute temperature (K), Δ*H*^0^ is the enthalpy (Δ*H*^0^, kJ mol^−1^), and Δ*S*^0^ is the entropy (Δ*S*^0^, J mol^−1^K^−1^). Δ*H*^0^ and Δ*S*^0^ are determined from the slope and the intercept of the linear fit of the van’t Hoff plot, e.g., ln *K_L_* versus 1/*T* (Figure 10). The calculated values are presented in Table 4.

The negative value of Δ*G*^0^ indicated that the adsorption process of Sb (III) on ox-xGnP was spontaneous and thermodynamically favored, the adsorbate having a high affinity for the adsorbent. The negative value of Δ*H*^0^ indicated that the process was exothermic. That was also confirmed by the increase of *K_L_* with temperature. The positive value of Δ*S*^0^ obtained from the adsorption process data suggested randomness at the solid–solution interface during the adsorption of Sb (III) on ox-xGnP and also the possibility of structural changes or readjustments in the adsorbate–adsorbent complex [41,42].

### 3.6. Proposed Mechanism for Adorption of Sb (III) onto ox-xGnP

To determine the adsorption mechanism, the following results of the studies were considered:(1)The adsorption model was Langmuir-type, which means that the active centers on the ox-xGnP surface were the same, evenly distributed and did not influence each other;(2)On the basis of literature data [43], the active species in the solution at pH = 7 was antimonous acid, Sb(OH)_3_ [32], and the active positions on the ox-xGnP surface were –COO, as emphasized by the FT-IR analysis (Figure 2).

The mechanism (Figure 11) was formulated as follows:

The uniformity of distribution of active sites on the ox-xGnP surface was demonstrated by the SEM image (Figure 3b). The fact that the active species of Sb (III) was Sb(OH)_3_ was supported by both the STEM and EDX analyses on one hand (Figure 4b,c), which revealed the presence of Sb, and the negative zeta potential values and increased adsorption capacity at higher pH on the other hand. Stabilization of the tetrahedral adsorbed ion was visible in the SEM image (Figure 3d).

## 4. Conclusions

In this paper, Sb (III) adsorption was evaluated by using ox-xGnP as an adsorbent. The ox-xGnP was characterized by TGA, FT-IR, BET, SEM, and TEM analysis techniques. To optimize the adsorption process, several parameters were studied, namely, the influence of pH, the influence of the ox-xGnP concentration, the influence of the contact time, the influence of the initial concentration of Sb (III), and the temperature. The study of these parameters revealed the best adsorption capacity of Sb (III) on ox-xGnP at pH = 7.0, with 1.0 mg ox-xGnP/100 mL solution, T = 293 K, 1.0 mg L^−1^ Sb (III), contact time 25 min. The results of Sb (III) adsorption on ox-xGnP showed that the best correlation of kinetic data was described by the pseudo-first-order kinetic model. The adsorption isotherms of Sb (III) onto ox-xGnP were better described by the Langmuir isotherm model than by the Freundlich isotherm. The maximum adsorption capacity obtained from the Langmuir isotherm was 18.2 mg g^−1^. The thermodynamic parameters studied showed that the adsorption process of Sb (III) on ox-xGnP was exothermic and spontaneous.

## Figures and Tables

**Figure 1 nanomaterials-08-00992-f001:**
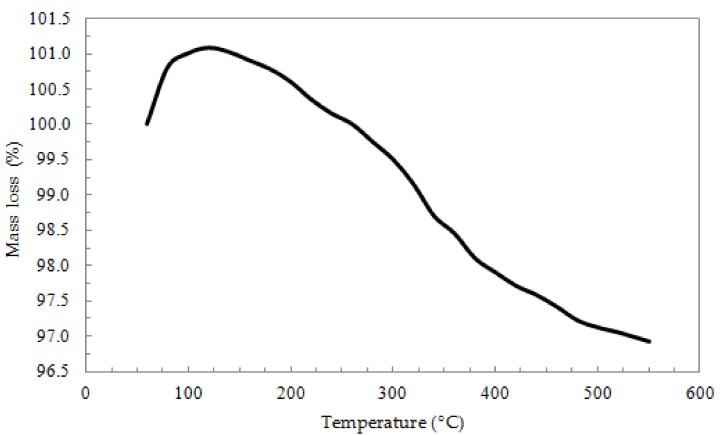
TGA of oxidized exfoliated graphite nanoplatelets (ox-xGnP).

**Figure 2 nanomaterials-08-00992-f002:**
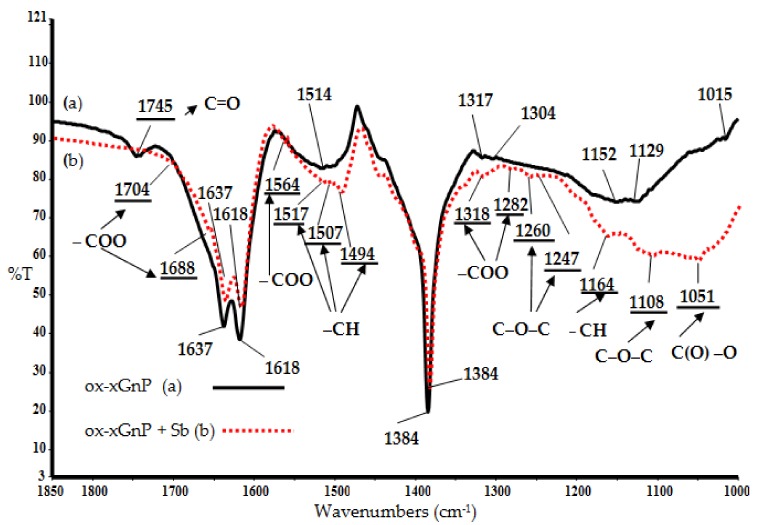
FT-IR spectra (**a**) ox-xGnP before adsorption of Sb (III); (**b**) ox-xGnP after adsorption of Sb (III) (ox-xGnP+Sb).

**Figure 3 nanomaterials-08-00992-f003:**
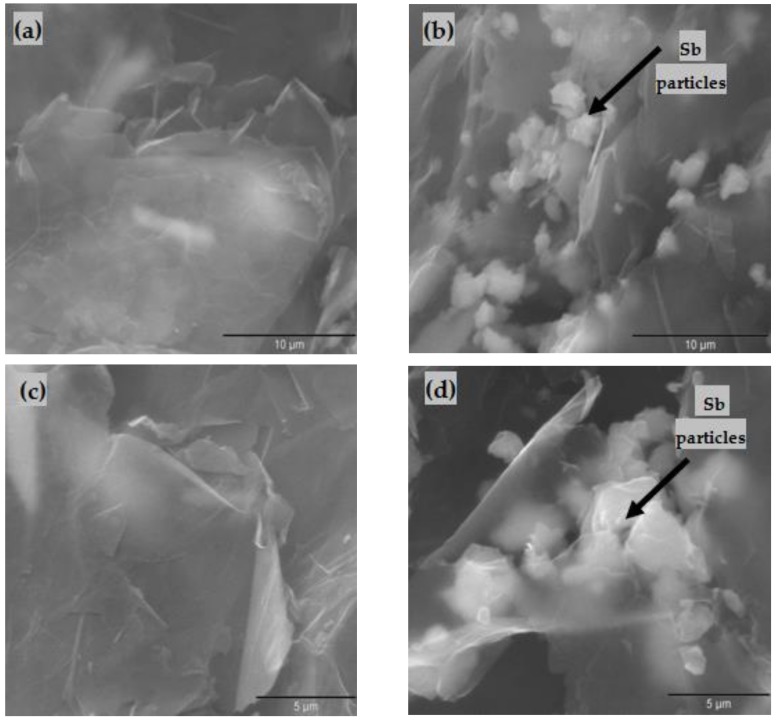
SEM images (**a**,**c**) ox-xGnP before adsorption of Sb at magnifications of 10,000× and 15,000×, respectively; (**b**,**d**) ox-xGnP after adsorption of Sb at magnifications of 10,000× and 15,000×, respectively.

**Figure 4 nanomaterials-08-00992-f004:**
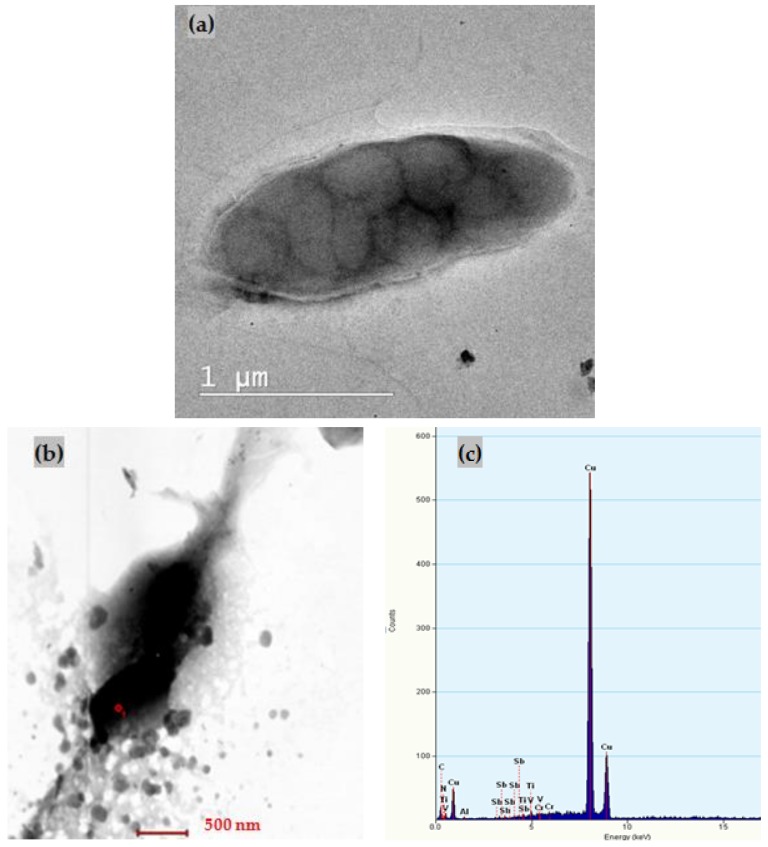
(**a**) TEM micrograph of ox-xGnP before adsorption of Sb; (**b**) STEM image of ox-xGnP after adsorption of Sb; (**c**) EDX analysis of ox-xGnP after adsorption of Sb.

**Figure 5 nanomaterials-08-00992-f005:**
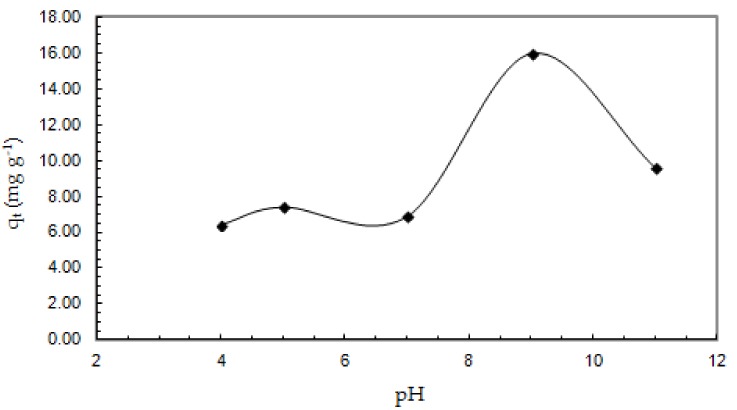
The effect of solution pH on the adsorption capacity of Sb (III) onto ox-xGnP (0.7 mg L^−1^ Sb (III), 1.0 mg ox-xGnP/100 mL solutions, T = 20 °C and contact time 30 min).

**Figure 6 nanomaterials-08-00992-f006:**
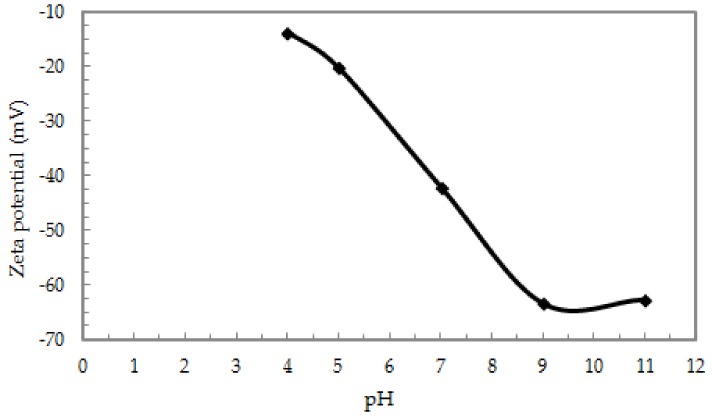
Zeta potential curve of ox-xGnP suspension versus solution pH.

**Figure 7 nanomaterials-08-00992-f007:**
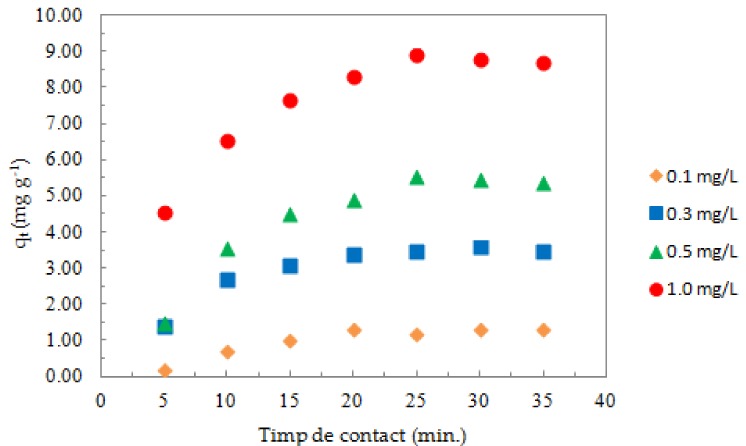
The effect of the initial Sb (III) concentration on its sorption on ox-xGnP (pH 7.0, 1.0 mg of ox-xGnP/100 mL solution, T = 20 °C).

**Figure 8 nanomaterials-08-00992-f008:**
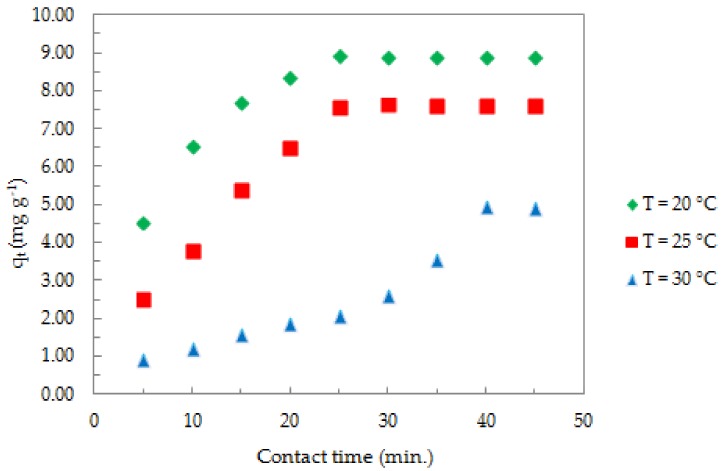
Effect of contact time and temperature on the adsorption of Sb (III) onto ox-xGnP (pH = 7.0, 1.0 mg L^−1^ Sb (III), 1.0 mg of ox-xGnP/100 mL solution).

**Figure 9 nanomaterials-08-00992-f009:**
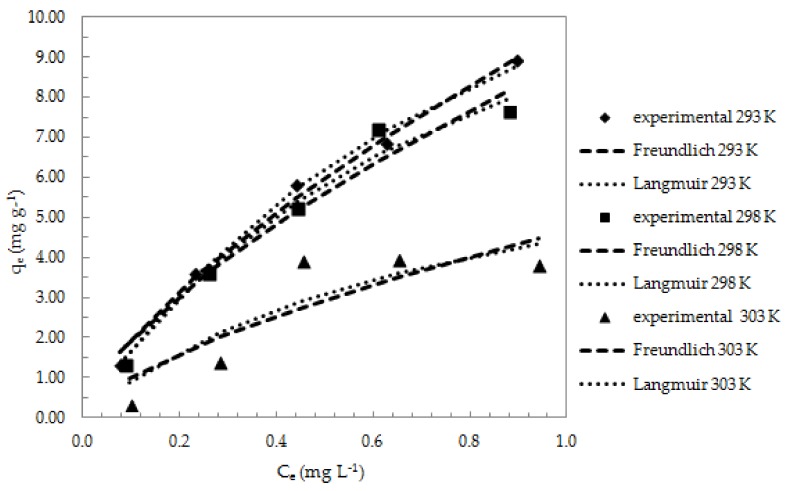
Non-linear Freundlich and Langmuir isotherms of Sb (III) adsorption on ox-xGnP (pH = 7.0, 1 mg L^−1^ Sb (III), 1 mg ox-xGnP/100 mL solution, T = 293 K, 298 K, 303 K, contact time 25 min).

**Figure 10 nanomaterials-08-00992-f010:**
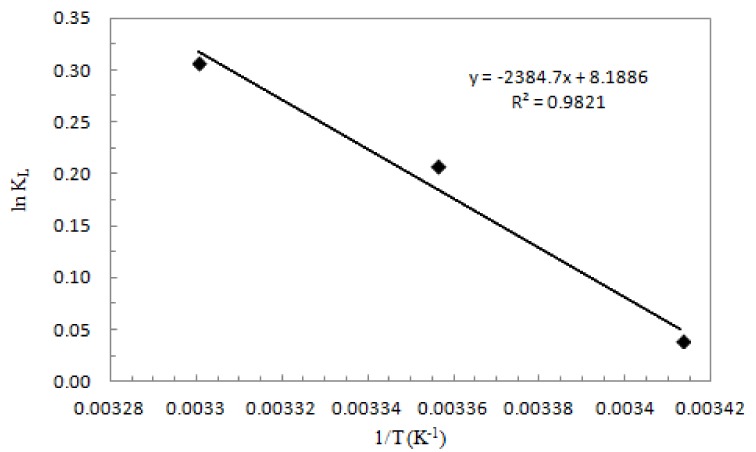
van’t Hoff plot for the adsorption of Sb (III) onto ox-xGnP.

**Figure 11 nanomaterials-08-00992-f011:**
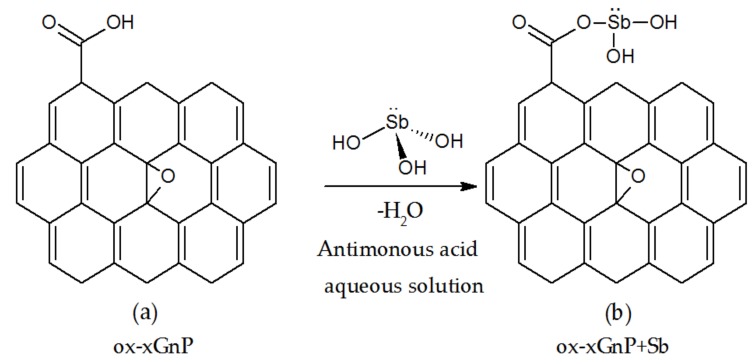
Proposed mechanism for adsorption of Sb (III) on ox-xGnP, (**a**) ox-xGnP, (**b**) ox-xGnP + Sb.

**Table 1 nanomaterials-08-00992-t001:** Effect of sorbent amount on the sorption capacity.

Amount of ox-xGnP (mg)	Concentration of Sb (III) (mg L^−1^)	Qe (mg g^−1^)
1.0	0.3	3.60
1.0	0.7	6.86
2.0	0.3	2.85
2.0	0.7	5.10

**Table 2 nanomaterials-08-00992-t002:** Parameters obtained for different kinetic models of the adsorption of Sb (III) on ox-xGnP, at T = 293 K.

*q_e_*_exp._ (mg g^−1^)	Pseudo-First-Order Kinetic Model			Pseudo-Second-Order Kinetic Model			Intra-Particle Diffusion Model		
	*q_e_*_cal._ (mg g^−1^)	*k*_1_ (min^−1^)	*R* ^2^	*q_e_*_cal._ (mg g^−1^)	*k*_2_ (g mg^−1^ min^−1^)	*R* ^2^	*k_id_* (mg g^−1^ min^−1/2^)	*C* (mg g^−1^)	*R* ^2^
8.91	8.10	0.129	0.999	11.3	199	0.998	1.64	1.17	0.978

**Table 3 nanomaterials-08-00992-t003:** Parameters of the isotherms for the adsorption of Sb (III) on ox-xGnP.

T (K)	Freundlich Isotherm				Langmuir Isotherm			
	*K_F_*	*n*	1/*n*	*R* ^2^	*K_L_* (L mg^−1^)	*q_m_* (mg g^−1^)	*R* ^2^	*R_L_*
T = 293	9.66	1.45	0.69	0.965	1.04	18.2	0.994	0.49
T = 298	8.88	1.51	0.66	0.965	1.23	15.3	0.989	0.45
T = 303	4.66	1.50	0.66	0.849	1.36	7.65	0.923	0.42

**Table 4 nanomaterials-08-00992-t004:** Thermodynamic parameters of Sb (III) adsorption on ox-xGnP. Δ*H*^0^: enthalpy, Δ*S*^0^: entropy, Δ*G*^0^: Gibb’s free energy.

ΔH^0^ (kJ mol^−1^)	ΔS^0^ (kJ mol^−1^ K^−1^)	ΔG^0^ (kJ mol^−1^)		
		T = 293 (K)	T = 298 (K)	T = 303 (K)
−19.8	0.068	−39.8	−40.1	−40.5

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
