# Peer review of "Adsorption of Sb (III) on Oxidized Exfoliated Graphite Nanoplatelets"

_nanomaterials, 2018, doi:10.3390/nano8120992_

Round 1

Reviewer 1 Report

Dear Editor,

The manuscript (nanomaterials-390819) is about employment of Graphene oxide nanoplatelets for removal (adsorption) of Sb(III) from water. The authors have extensively characterized this system from different perspectives of chemical, morphological, thermal etc. I give credit to the authors for such wide evaluations, but there are some major concerns that need to be addressed. My major comments are:

1- Introduction, Page 2, line 53-54 “Few research on the adsorption of …” This statement needs more references than only one.

2- In general, Figures have been presented in a very primitive way. Frames, fonts, style, quality etc. need to be largely modified. This problem applies for SEM images, as well. The lower margin can be cropped, instead a scale bar added. Sb particles should be marked.

3- Figure 1, TGA graph, what rise and decline of mass (%) with temperature mean? This result should be discussed.

4-   Page 4, line 142, how pore in this material is defined, interplanar or …? A schematic of the material (structure) could help.

5- Page 4, line 143, 3.89 nm should be rounded up. Pore volume means porosity?

6- what Fig4a and 4b are supposed to show? No justification, explanation, discussion?

7- Figure 8, effect of temperature on Sb adsorption, only 5 ÂşC difference can make such significant changes in adsorption amount? I was wondering if the authors have repeated the test? I remember that they mentioned the experiments have been repeated three times, if this is the case, please add standard deviations.

8- Page 9, lines 237-238, increase of temperature leads to decline of adsorption efficiency. Can the authors discuss this behavior?

9- Page 14, Table 5, due to different experimental (working) conditions of the referred citations, it is very hard to compare the systems and introduce a system as a superior one. I suggest the authors to remove the table.

10- Table 5, GO has been introduced as a counterpart (ref [40]) for the adsorbent studied here. I assume the system being studied in the article is also GO. If this is right, where is novelty of the current work? While, the authors in first sentence of abstract state:” In this work, Sb (III) adsorption on oxidized exfoliated graphene nanoplatelets (ox-xGnP) was evaluated for the first time to the best of our knowledge”.

Author Response

 Hello,

 Please see my document attached here.

 Best regards,

 Luiza Capra

Reviewer 2 Report

Journal:  Nanomaterials

Manuscript Number: nanomaterials-390819

Title:   Adsorption of Sb (III) on oxidized exfoliated 2 graphene nanoplatelets

Author(s):    Luiza Capra, Mihaela Manolache, Ion Ion, Rusandica Stoica, Gabriela Stinga, Sanda Maria Doncea, Elvira Alexandrescu, Raluca Somoghi, Marian Romeo Calin, Ileana Radulescu, Georgeta Ramona Ivan, Marian Deaconu, Alina Catrinel Ion

Reviewer Comments to the Editor and Authors

General comments:

This paper is well organized and readable, which is also within the Journal’s scope. However, I want to spotlight that although authors did a lot of work especially on ox-xGnP characterization, this paper presents a significant number of fundamental drawbacks:

Major comments:

1.       Adsorption efficiency

Adsorbents’ efficiency for pollutants removal will be estimated under conditions similar to those found either in water or wastewater samples. An important task is the proper evaluation of obtained results with respect to the compliance to the maximum contaminant limit (MCL) for each pollutant. Unfortunately, in this study, as well as in the majority of bench-scale experiments, the efficiency of adsorbents is judged through the maximum adsorption capacity (Qmax), which defined by the plateau of the recorded isotherm observed at extremely high residual concentrations (see curve A in Figure A). In some other cases, the percentage of initial pollutant concentration decrease after treatment is given as a criterion for the successful adsorbent’s performance (see curve C in Figure A). The weakness of both

Figure A. Simplified approach to evaluate applicability of adsorbents in water purification by adsorption isotherms. Case A: material succeeding high maximum capacity but completely fail to decrease pollutant concentration below the MCL. Case B: high performing material in the concentration range of the MCL; effective operational capacity is signified at the section of the curve with the MCL. Case C is an example of mistaken use of percentage removal for the evaluation of adsorbents independently to the residual concentration. Point 1 indicates the removal of 95 % of an unrealistic initial pollutant’s concentration (e.g. >10 mg/L) succeeding a high residual concentration. Point 2 corresponds to the removal of 70 % of a realistic initial pollutant’s concentration (e.g. 40 μg/L) reaching residual concentration below the MCL.

approaches is that they usually point to high initial and residual concentrations which indeed bring high adsorption capacities and percentage removals but provide no data for the ability to reach low concentrations such as the regulation limits.

Depending on the pollutant removal mechanism, the risk of using such evaluation criteria is related to the possible failure of specific adsorbents to reduce concentration below legislation demands although they achieve very high maximum adsorption capacities. For this reason, a better way to monitor the efficiency of various adsorbents is by the introduction of an index defined after the adsorption capacity that corresponds to a residual concentration equal to the regulation limit of each pollutant (QMCL). In practice, this index directly indicates the operational capacity and lifetime of the material and it is estimated by the projection of adsorption isotherm to the MCL of the studied heavy metal (see curve B in Figure A). For example, the evaluation index for the adsorbent used in this study to remove Sb(III) should be determined by the adsorption capacity from isotherms at the residual concentration of 5 μg/L, which is the value of ΕU regulation limit for drinking water. The value of this index is expected to provide a good estimation of the adsorbent efficiency during the operation of a large-scale unit indicating the maximum operational capacity before its replacement. More specifically:

·         Abstract: The referred adsorption capacity 8.91 mg/g was determined at equilibrium concentration close to 800 ÎĽg/L!

·         Figure 12. The calculated adsorption capacities for natural water samples referred to residual concentration by far higher to drinking water regulation limit e.g.:

Ultrapure water                                                               Ce 631 ÎĽg/L !

Tap water                                                            Ce 638 ÎĽg/L !

Non-carbonated mineral water                 Ce 657 ÎĽg/L !

Carbonated mineral water                           Ce 658 ÎĽg/L !

Surface water                                                    Ce 663 ÎĽg/L !

·         The results of Table 5 make no sense because they do not referred to equal equilibrium (Ce) concentration.

 Conclusively, the main experimental results of the study presents no practical approximation since they do not clarify the ability of ox-xGnP to remove Sb(III) below drinking water regulation limit, as well as it adsorption capacity (q5)at Ce 5 ÎĽg/L. It is self-obvious that the comment “The adsorption capacity of Sb (III) on ox-xGnP was studied on four different samples of water and the results obtained show that ox-xGnP can be used to remove Sb (III) from different water samples. The advances obtained in this work will help find new nanosorbents applied in natural environments in order to remove inorganic aqueous contaminants” (page 16 – lines 429-433) is speculation.

2.       Adsorption evaluation

Sections 3.2.2 and 3.2.3

·   The increase of adsorbent dosage in a solution with constant Sb(III) concentration (0.3 or 0.7 ÎĽg/L) results of course in lower equilibrium concentration, which according to adsorption models (i.e. q=KCen) results in lower adsorption capacity (Table 1). So the comment (page 8, line 212-213) is incorrect.

Table 1. Effect of sorbent amount on the sorption capacity.

The amount of

ox-xGnP (mg)

Concentration

of Sb (III) (mg/L)

qe , (mg/g)

Ce, ÎĽg/L

1

0.3

3.60

264

1

0.7

6.86

631

2

0.3

0.7     wrong value

2

0.7

5.10

598

·   Similarly, the increase of initial concentration of Sb(III), while keeping the adsorbent dose constant (1 ÎĽg/L), results in higher equilibrium concentration which according to adsorption model results in greater adsorption capacity. So the comment (page 8, line 222) is incorrect.

·   So, Table 1 and Figure 7 incorporate a wrong presentation of adsorption data!!

·   Conclusively, politely speaking the presentation of the results (make no sense) is at least incorrect.

3.       Proposed mechanism for adsorption of Sb(III) onto ox-xGnP

Figure 1 presents the Sb(III) speciation at low concentration and drinking water matrix. In addition experimental results with iron oxy-hydroxides showed that adsorption efficiency favoured by positive charge density, while the concentration of Sb(OH)2+ specie in pH range 6-8 is zero. So, it is difficult to accept the suggested adsorption mechanism which does not supported by EXAFS data.

Figure 1. Percentage of Sb(III) species at concentrations 100 ÎĽg/L in tap water matrix and 20o C. Diagrams derived by Visual MINTEQ 3.0 (http://vminteq.lwr.kth.se).

Minor comments

Abstract

The adsorption parameters, such as pH and contact time were optimized and the best adsorption  capacity obtained was 8.91 mg g-1.

Clarify the optimum conditions

Introduction

 Due to the fact that water is distributed and stored in bottles (polyethylene terephthalate), it can be contaminated with 38 various heavy metals (Cr, Cd, Co, Cu, As, Hg, Pb, Ni, Sb) [1].

Delete this comment

Kinetics

The data of kinetics referred to equilibrium concentrations greater than 600 ÎĽg/L. I think that do not represent the contact time for equilibrium concentrations close to drinking water regulation limit.

Conclusions

Should be modified according to above mentioned comments.

Conclusively, this paper although presents fundamental technological drawbacks should be accepted only after improve and drastic modification of experimental results. To help authors I suggest to read (not to cited) the paper: “Efficiency of iron-based oxy-hydroxides in removing antimony from groundwater to levels below the drinking water regulation limits”, Sustainability Journal, 9, 238, 1-11, 2017, doi: 10.3390/su9020238.

Author Response

(The authors gave the same response as above.)

Round 2

Reviewer 1 Report

Dear Editor,

My main concerns have been properly addressed. The paper is now publishable.

Reviewer 2 Report

The paper was significantly improved according to reviewers’ comments and should be accepted for publication.